

# Genome-wide identification of CBL family and expression analysis of *CBLs* in response to potassium deficiency in cotton

Tingting Lu[1,2,*], Gaofeng Zhang[1,*], Lirong Sun[1], Ji Wang[1] and Fushun Hao[1]

[1] State Key Laboratory of Cotton Biology, Henan Key Laboratory of Plant Stress Biology, College of Life Sciences, Henan University, Kaifeng, Henan, China

[2] College of Pharmaceutical Engineering, Henan University of Animal Husbandry and Economy, Zhengzhou, Henan, China

[*] These authors contributed equally to this work.

## ABSTRACT

Calcineurin B-like (CBL) proteins, as calcium sensors, play pivotal roles in plant responses to diverse abiotic stresses and in growth and development through interaction with CBL-interacting protein kinases (CIPKs). However, knowledge about functions and evolution of CBLs in *Gossypium* plants is scarce. Here, we conducted a genome-wide survey and identified 13, 13 and 22 CBL genes in the progenitor diploid *Gossypium arboreum* and *Gossypium raimondii*, and the cultivated allotetraploid *Gossypium hirsutum*, respectively. Analysis of physical properties, chromosomal locations, conserved domains and phylogeny indicated rather conserved nature of CBLs among the three *Gossypium* species. Moreover, these CBLs have closer genetic evolutionary relationship with the CBLs from cocoa than with those from other plants. Most CBL genes underwent evolution under purifying selection in the three *Gossypium* plants. Additionally, nearly all *G. hirsutum* CBL (GhCBL) genes were expressed in the root, stem, leaf, flower and fiber. Many *GhCBLs* were preferentially expressed in the flower while several *GhCBLs* were mainly expressed in roots. Expression patterns of GhCBL genes in response to potassium deficiency were also studied. The expression of most *GhCBLs* were moderately induced in roots after treatments with low-potassium stress. Yeast two-hybrid experiments indicated that GhCBL1-2, GhCBL1-3, GhCBL4-4, GhCBL8, GhCBL9 and GhCBL10-3 interacted with GhCIPK23, respectively. Our results provided a comprehensive view of the *CBLs* and valuable information for researchers to further investigate the roles and functional mechanisms of the CBLs in *Gossypium*.

## INTRODUCTION

Calcium ion ($Ca^{2+}$) plays pivotal roles in mediating and regulating many fundamental growth and developmental processes and in response to various environmental stimuli (*Luan, 2009*; *Kudla, Batistič & Hashimoto, 2010*; *Sarwat et al., 2013*). The $Ca^{2+}$ signals are primarily perceived by some $Ca^{2+}$ sensors including $Ca^{2+}$ dependent protein kinases,

Corresponding author
Fushun Hao, haofsh@henu.edu.cn

calmodulins and calcineurin B-like proteins (CBLs), and then are transmitted by these sensors to downstream targets to initiate diverse cellular responses (*Luan, 2009*; *Kudla, Batistič & Hashimoto, 2010*; *Sarwat et al., 2013*).

CBLs are proteins sharing sequence similarity with the B subunit of calcineurin B in yeast and neuronal calcium sensors in animals (*Kudla et al., 1999*). Each CBL has at least three EF domains and $Ca^{2+}$-binding sites (*Mohanta et al., 2015*; *Mao et al., 2016*). CBLs relay $Ca^{2+}$ signals through interaction with and activation of the CBL-interacting protein kinases (CIPKs). Moreover, CBL-CIPK has been demonstrated to serve as an essential signaling network regulating plant responses to multiple abiotic stresses such as salinity, $K^+$ deficiency, excess of $Mg^{2+}$ and drought (*Sanyal, Pandey & Pandey, 2015*; *Thoday-Kennedy, Jacobs & Roy, 2015*; *Mao et al., 2016*). It also modulates growth and development, absorption and/or transport of nitrate, ammonium and iron, sustaining of $H^+$ homeostasis, and transduction of reactive oxygen species signals in plants (*Sanyal, Pandey & Pandey, 2015*; *Thoday-Kennedy, Jacobs & Roy, 2015*; *Mao et al., 2016*).

In *Arabidopsis*, 10 genes (*CBL1-10*) encoding CBL proteins have been found (*Kolukisaoglu et al., 2004*). *CBL1* and *CBL9* were reported to positively regulate the uptake and transport of $K^+$, $NO_3^-$, $NH_4^+$, aluminum and iron, and the promotion of stomatal opening (*Li et al., 2006*; *Xu et al., 2006*; *Ho et al., 2009*; *Mao et al., 2016*; *Tian et al., 2016*; *Ligaba-Osena et al., 2017*; *Straub, Ludewig & Neuhäuser, 2017*). *CBL1* and *CBL9* also affect abscisic acid (ABA)-induced stomatal closure and ROS signaling (*Pandey et al., 2004*; *Cheong et al., 2007*; *Drerup et al., 2013*). *CBL2* plays a negative role in the activation of plasma membrane (PM) $H^+$-ATPase (*Fuglsang et al., 2007*). Moreover, *CBL2* and *CBL3* are cooperatively implicated in sequestering $Mg^{2+}$ and modulation of pollen germination and tube growth (*Steinhorst et al., 2015*; *Tang et al., 2015*). *CBL3* are also engaged in $K^+$ distribution and translocation (*Liu et al., 2013*). *CBL4* was proven to be a crucial regulator for excluding $Na^+$ and translocation of AKT2 (*Arabidopsis* $K^+$ transporter 2) from endoplasmic reticulum to PM (*Held et al., 2011*). *CBL10* is involved in enhancing salt tolerance, stimulating $K^+$ absorption, and modulating GTPase activity (*Kim et al., 2007*; *Ren et al., 2013*; *Cho et al., 2016*). In cotton (*Gossypium hirsutum*), *GhCBL2* and *GhCBL3* appear to modulate fiber elongation (*Gao et al., 2008*). Many *CBLs* in other plant species also play important parts in regulating the responses to various abiotic stress as well as growth and development (*Li et al., 2014a*; *Thoday-Kennedy, Jacobs & Roy, 2015*).

In recent years, multiple CBL gene families have been identified at genome-wide levels in rice, maize, wheat and other plants (*Kolukisaoglu et al., 2004*; *Zhang et al., 2014*; *Sun et al., 2015*; *Li et al., 2016a*; *Li et al., 2016b*; *Zhang et al., 2016*). Some conserved domains such as EF-hands, myristoylation and palmitoylation sites were discovered in CBLs (*Kolukisaoglu et al., 2004*; *Mohanta et al., 2015*). The expression patterns of many *CBL* genes were also investigated in different tissues and in response to various abiotic stresses in plants (*Mohanta et al., 2015*; *Zhang et al., 2016*). These findings lay the foundation for people to further explore the functional mechanisms of CBLs in plants. However, to date, knowledge about genomics and evolutionary information of CBLs in *Gossypium* is limited.

Cotton is an essential tetraploid fiber crop that supplies lint for the textile industry worldwide. It is considered to descend from an ancestral combination of two diploid most

similar to modern A (for example *Gossypium arboretum*) and D genome species (*Gossypium raimondii*) (*Wendel, Brubaker & Seelanan, 2010*).

Cotton growth and development are severely threatened by diverse abiotic stresses such as drought, salinity and potassium starvation (*Allen, 2010*). Therefore, enhancing stress tolerance of cotton cultivars is one of most important strategies for us to improve their productivity and quality. Potassium is a vital macronutrient for plants, especially for cotton. Potassium shortage in soil seriously affects the yield and quality of cotton (*Oosterhuis, Loka & Raper, 2013*). Moreover, it has been demonstrated that $K^+$ uptake is controlled by CBLs through interacting with CIPK23 in *Arabidopsis* and rice under potassium deficiency (*Li et al., 2014a*; *Mao et al., 2016*). Research is needed to determine which and how CBLs modulate $K^+$ absorption in cotton. In this report, genome-wide and comprehensive analyses of the CBL family in *G. arboreum*, *G. raimondii* and *G. hirsutum* were conducted. The expression patterns of *GhCBLs* were monitored in tissues and in response to potassium deficiency in cotton. These analyses will provide a basis for further investigation of the functions of CBLs in *Gossypium*.

## MATERIALS AND METHODS

### Identification of CBL family in *Gossypium*

The genome sequences of *G. arboreum* (BGI-CGB v2.0 assembly genome), *G. raimondii* (JGI assembly v2.0 data.) and *G. hirsutum* (NAU-NBI v1.1 assembly genome) were downloaded from the CottonGen database (www.cottongen.org), respectively. The protein sequences of 10 *Arabidopsis* CBLs were applied as queries to search the three genomes using BLAST-2.4.0 software (ftp://ftp.ncbi.nlm.nih.gov/blast/executables/blast+/LATEST) with default parameters ($E$-value $<$ $e^{-10}$). EF-hand domains, the typical CBL domains, were analyzed within the candidate CBLs one by one using online software SMART (http://smart.embl-heidelberg.de/). The CBL motifs were also queried against the Pfam databases (*Finn et al., 2010*). The putative CBLs with questionable annotations (i.e., having a typical CBL domain but low $E$-value or low coverage of a domain) were manually reanalyzed.

### Analysis of *Gossypium CBLs* family

The properties of the *Gossypium* CBL proteins were analyzed using online tools ExPaSy (http://web.expasy.org/protparam/). The subcellular localizations of the CBLs were examined in the website http://www.csbio.sjtu.edu.cn/bioinf/Cell-PLoc/. The locations of the *CBLs* in chromosomes were assessed by MapInspect software (http://www.softsea.com/review/MapInspect.html). Structures of the CBLs were determined by GSDS (http://gsds.cbi.pku.edu.cn/). The conserved domains in the CBLs were affirmed by SMART (http://smart.embl-heidelberg.de). The sequence logo of myristoylation motif in the CBLs was generated by MEME program (http://meme-suite.org/tools/meme).

### Analyses of synteny and Ka/Ks ratio

The homologous gene pairs among the *Gossypium* CBLs were searched by the MCScanx software (http://chibba.pgml.uga.edu/mcscan2/). The gene collinearity results were

obtained by CIRCOS program (http://www.circos.ca/). The ratio of Ka (nonsynonymous substitution rate) to Ks (synonymous substitution rate) of the CBL genes were estimated by PAML program (http://abacus.gene.ucl.ac.uk/software/paml.html).

## Phylogenetic analysis of CBLs

The CBL data were downloaded from the websites for various plant species including *Arabidopsis thaliana* (http://www.arabidopsis.org/), *Oryza sativa* (http://rapdb.dna.affrc.go.jp), *Vitis vinifera* (http://www.genoscope.cns.fr/spip/Vitis-vinifera-e.html), *Populus trichocarpa* (http://www.phytozome.net/poplar), *Glycine max* (http://www.phytozome.net/soybean), *Theobroma cacao* (http://cocoagendb.cirad.fr), *Carica papaya* (http://www.hawaii.edu/microbiology/asgpb/) and castor bean (http://castorbean.jcvi.org). The full-length amino acid sequences of CBL proteins were aligned using Clustal W software through pairwise and multiple alignment with default parameters (*Larkin et al., 2007*). Then, phylogenetic trees were generated based on the alignment results using the neighbor joining method (Neighbor-Joining, NJ) and 1,000 bootstrap trials with the MEGA 5.0 software (http://www.megasoftware.net/).

## Expression analysis of *GhCBL* genes in tissues and in response to potassium deficiency

For measuring the expression of the *GhCBLs* in tissues, samples of roots, stems and leaves were collected from 20-day-old *G. hirsutum* TM-1 plants normally grown in soil containing 1:1 (v:v) peat:vermiculite in a growth chamber (day/night temperature cycle of 28 °C/26 °C, 14 h light/10 h dark, and about 50% relative humidity). Flowers were isolated in the morning at the first day of anthesis from cotton grown in the field. The fibers at elongation stage were obtained from the ovules (23 days post anthesis). For monitoring the expression of *GhCBLs* in responding to potassium deprivation, cotton plants grew in clean small pebbles (watered by liquid 1/2 MS medium) (*Murashige & Skoog, 1962*) in the growth chamber described above for 3 weeks. Then, the plants were watered with $K^+$-lacking liquid 1/2 MS medium ($KNO_3$ was replaced by $NH_4NO_3$ and $KH_2PO_4$ was replaced by $NH_4H_2PO_4$) for 0 h, 6 h, 2 d and 5 d, respectively. Meanwhile, some $K^+$-starved seedlings for 5 d were resupplied with $K^+$ (watered with $K^+$-contained 1/2 MS medium) for 3 h. The cotton roots were collected, immediately frozen in liquid nitrogen and stored at $-70$ °C. Total RNA of samples was extracted using RNA Pure Plant Kit's protocol (Tiangen Biotech, Beijing, China). The purity of RNA was examined using a Nanodrop2000 nucleic acid analyzer. The A260/280 ratio for each RNA sample was about 2.0. Then, total cDNA was synthesized using M-MLV reserve transcriptase synthesis system (Promega, Madison, WI, USA) following the instructions in the Promega kit (https://tools.thermofisher.com/content/sfs/manuals/superscriptIII_man.pdf).

Quantitative real-time RT-PCR (qRT-PCR) experiments were performed using the cDNA, SYBR Green Master mix, the specific primers of *GhCBL* genes (Table 1), and an ABI 7500 real-time PCR system. *GhUBQ7* was used as the internal control. At least three biological replicates were carried out.
**Table 1** Gene primers used for quantitative real-time RT-PCR experiments.

| Genes | AGI number | Forward primers (5′–3′) | Reverse primers (5′–3′) |
| --- | --- | --- | --- |
| GhUBQ7 | Gh_A11G0969 | GAAGGCATTCCACCTGACCAAC | CTTGACCTTCTTCTTCTTGTGCTTG |
| GhCBL1-1 | Gh_A11G0257 | GAGCGTAACGAGGTCAAGCAAA | CTTCCCGTCCTGATTAATGTCC |
| GhCBL1-2 | Gh_D11G0276 | TTTTGTTCGAGCACTCAATGTTT | TTGCCTCAATCGTTTCATCAG |
| GhCBL1-3 | Gh_A03G0043 | GACATTCTTGGAAGCCGATA | CTGAGGTATGGGAGGGTCAT |
| GhCBL1-4 | Gh_D09G1875 | AGAGTAATGACCCTCCCATACCTAA | CGAGCGAGTATTCTCCGACAA |
| GhCBL1-5 | Gh_A09G1766 | GGATGCCGACACTAACCAGG | TCCAACAACGTAGCGGCC |
| GhCBL3-1 | Gh_A01G0740 | AGTTTGCTCGTGCTCTCTGT | ATCATCTGAAAGGTTCATGCCA |
| GhCBL3-2 | Gh_D01G0760 | GCAAGAGAGACCGTTTTTAGTG | AATCTTATCGTCAATGGGCG |
| GhCBL3-3 | Gh_A13G1099 | GGGCTGATTAACAAGGAGGAGT | ACAGAAAGAGCACGAGCAAACT |
| GhCBL3-4 | Gh_D13G1364 | ATGGGCTGATTAACAAGGAGGAG | GACAGAAAGAGCACGAGCGAAC |
| GhCBL3-5 | Gh_A04G0051 | GCGGTGATAGATGACGGACT | GACAGAGAGAGCACGAGCAA |
| GhCBL3-6 | Gh_D05G3682 | TACACGCTTCCGACCCTATT | ATCAATGAGCCCGTCGTAAC |
| GhCBL4-1 | Gh_A11G0126 | ACGGCTAGTGAAGTAGAATCCC | CGAACAAATCAAAAACCCTGTC |
| GhCBL4-2 | Gh_D11G0140 | TTCTTGCTGCTGAAACACCT | CGAACAAATCAAAAACCCTG |
| GhCBL4-3 | Gh_A12G2144 | TAAGCGTCTTTCATCCCAAC | TGATTCACCAAGCAGAGCCA |
| GhCBL4-4 | Gh_A09G1696 | AACTTAGACACAAGGCTGGGTATG | GAGGTTCTGCTTATTGCTGTTTTT |
| GhCBL4-5 | Gh_D12G2320 | CCTGAGGAGGTCAAGGAGATG | AAATTGGGTTGCGAGCTACAAA |
| GhCBL9 | Gh_D08G1764 | GACATTCTTGGATGCCGACA | ACGCAGCAACCTCGTCTACT |
| GhCBL10-1 | Gh_A06G0800 | AGTCTCACAGTGGCGGCA | TTCATTGGCAAGACGGGTAA |
| GhCBL10-2 | Gh_D06G0922 | GTCGCGAGAAATGCCGTTAT | ATTCTCGCCGTATGGAGTTTG |
| GhCBL10-3 | Gh_A05G0335 | CTGAAATGAATTTGTCCGATGAC | ACTGGAAATAGTAGTTCATCACGGA |
| GhCBL10-4 | Gh_D05G0440 | TCTGGAATGAATTTGTCGGATG | CTGGAAATAGGAGTTCTTCACGG |

## Yeast two-hybrid (Y2H) analysis

The full-length CDS sequences of *GhCBLs* and *GhCIPK23* genes were amplified, sequenced and cloned into pGBKT7 and pGADT7 vectors, respectively, using primers listed in Table 2. The plasmids were then transformed into yeast strain AH109 according to the method described in page 18–21 in Yeast Protocols Handbook (Clontech, http://www.clontech.com/xxclt_searchResults.jsp). The cotransformants were plated on non-selective SD/-Leu/-Trp (synthetic dropout medium without Leu and Trp) solid medium and selective SD/-Leu/-Trp/-His/-Ade solid medium. The medium was prepared by ourselves. The concentrations of each component for SD/-Leu/-Trp medium are as follows: L-isoleucine 300 mg/L, L-valine 1.5 g/L, adenine 200 mg/L, L-arginine 200 mg/L, L-lysine 300 mg/L, L-methionine 200 mg/L, L-phenylalanine 500 mg/L, L-threonine 2 g/L, L-tyrosine 300 mg/L, L-histidine 200 mg/L, uracil 200 mg/L, yeast nitrogen base without amino acids 6.7 g/L, glucose 20 g/L. Serial 1:10 dilutions of the cotransformants were made in water, and 2 μl of the dilution was dropped to generate one spot. Plates were incubated at 30 °C for 3-4 d. 5-bromo-4-chloro-3-indoxyl- α-D-galactopyranoside (X- α-Gal) staining assay was carried out following the instruction (the Clontech protocol, page 26).

Lu et al. (2017), *PeerJ*, DOI 10.7717/peerj.3653

**Table 2  Gene primers used for yeast two-hybrid experiments.**

| Genes | AGI number | Forward primers (5′–3′) | Reverse primers (5′–3′) |
|---|---|---|---|
| *GhCBL1-2(BD)* | Gh_D11G0276 | CCGGAATTCATGGGCTGCTTTCAATCT | CGCGGATCCTGTGGCAACCTCATCA |
| *GhCBL1-3(BD)* | Gh_A03G0043 | CCGGAATTCATGGGTTGCTTTCATTCT | CGCGGATCCAGTGGCAACTTCATCTAC |
| *GhCBL1-4(BD)* | Gh_D09G1875 | CGCGGATCCATGGGCTGCTTGCAATGTAAA | GCACTGCAGTATGCCATTCGCCGAGCGAGT |
| *GhCBL1-5(BD)* | Gh_A09G1766 | ATAGGATCCATGGGCTGCTTGCAATGTA | GCACTGCAGGTATAACATCGGTATTATGTACCT |
| *GhCBL3-2(BD)* | Gh_D01G0760 | CGCGGATCCATGTTGCAGTGCATAGAC | GCACTGCAGTGTATCATCAACTTGAGAGTGGAAAA |
| *GhCBL3-4(BD)* | Gh_D13G1364 | CGCGGATCCATGGGAATTTGTTGTTTT | GCACTGCAGTTTGCCACCCATATTCAACT |
| *GhCBL4-1(BD)* | Gh_A11G0126 | CGCGGATCCATGAAATGGTGTTTTCAAACT | GCACTGCAGATCTCCATTGACGGAGACGCT |
| *GhCBL4-3(BD)* | Gh_A12G2144 | CGCGGATCCATGGGTTGTTTTTGCTTG | GCACTGCAGCTTATTCCCAACGATTTCAGCT |
| *GhCBL4-4(BD)* | Gh_A09G1696 | CGCGGATCCATGGGCTGCTTTTGCTTG | GCACTGCAGGTTTTTTCTCAATTCTTCACTGGT |
| *GhCBL8(BD)* | Gh_D09G1801 | CGCGGATCCATGGGCTGCTTTTGCTTGAAGAA | GCACTGCAGATTCTTCACTGGTTGCTGCAAATCTGAGAC |
| *GhCBL9(BD)* | Gh_D08G1764 | CCGGAATTCATGGGCTGCTTTCATTCT | CGCGGATCCCGCAGCAACCTCGTCTA |
| *GhCBL10-3(BD)* | Gh_A05G0335 | CGCGGATCCATGGATTCAACTAGCAAAACC | GCACTGCAGCCGGAGATAGGAAAGGGCCAA |
| *GhCIPK23(AD)* | Gh_A06G1219 | CCGGAATTCATGGCGAATCGCACTAGT | CGCGGATCCACCATCCTTTTCTTCCAC |

**Table 3  The CBL family genes in *Gossypium*.**

| Gene name | Gene ID | pI | MW (kDa) | Hydrophilicity | Predicted subcellular localization | Amino acid residues | Coding sequence |
|---|---|---|---|---|---|---|---|
| GaCBL1-1 | Cotton_A_16036 | 4.74 | 24.33 | −0.163 | Cell membrane | 213 | 642 |
| GaCBL1-2 | Cotton_A_16034 | 4.74 | 24.33 | −0.163 | Cell membrane | 213 | 642 |
| GaCBL1-3 | Cotton_A_16590 | 5.06 | 25.39 | −0.216 | Cell membrane | 221 | 666 |
| GaCBL1-4 | Cotton_A_09151 | 4.72 | 24.39 | −0.142 | Cell membrane | 213 | 642 |
| GaCBL2 | Cotton_A_07469 | 4.78 | 25.94 | −0.2 | Cell membrane | 226 | 681 |
| GaCBL3-1 | Cotton_A_06492 | 4.77 | 25.98 | −0.189 | Cell membrane | 226 | 681 |
| GaCBL3-2 | Cotton_A_02147 | 5.08 | 27.68 | −0.314 | Cell membrane | 240 | 723 |
| GaCBL4-1 | Cotton_A_02388 | 4.81 | 24.88 | −0.13 | Cell membrane | 220 | 663 |
| GaCBL4-2 | Cotton_A_13237 | 4.97 | 24.47 | −0.173 | Cell membrane | 215 | 648 |
| GaCBL8 | Cotton_A_08153 | 4.89 | 23.48 | −0.134 | Cell membrane | 205 | 618 |
| GaCBL9 | Cotton_A_13238 | 4.65 | 24.22 | −0.141 | Cell membrane | 210 | 633 |
| GaCBL10-1 | Cotton_A_14000 | 4.55 | 23.25 | −0.175 | Cell membrane | 199 | 600 |
| GaCBL10-2 | Cotton_A_34841 | 4.82 | 32.43 | −0.028 | Cell membrane | 279 | 840 |
| GrCBL1-1 | Gorai.007G030300 | 4.72 | 24.38 | −0.143 | Cell membrane | 213 | 642 |
| GrCBL1-2 | Gorai.003G178700 | 4.71 | 24.45 | 0.075 | Cell membrane | 214 | 645 |
| GrCBL1-3 | Gorai.004G191400 | 4.67 | 23.86 | 0.016 | Cell membrane | 209 | 630 |
| GrCBL1-4 | Gorai.006G214700 | 4.99 | 25.39 | −0.226 | Cell membrane | 221 | 666 |
| GrCBL3-1 | Gorai.013G150400 | 4.79 | 25.96 | −0.208 | Cell membrane | 226 | 681 |
| GrCBL3-2 | Gorai.002G102900 | 4.77 | 25.98 | −0.189 | Cell membrane | 226 | 681 |
| GrCBL3-3 | Gorai.009G450400 | 4.84 | 23.25 | −0.21 | Cell membrane | 226 | 681 |
| GrCBL4-1 | Gorai.007G015400 | 4.78 | 24.91 | −0.193 | Cell membrane | 233 | 702 |
| GrCBL4-2 | Gorai.006G207100 | 4.98 | 25.26 | −0.161 | Cell membrane | 221 | 666 |
| GrCBL4-3 | Gorai.008G255900 | 5.11 | 24.02 | −0.161 | Cell membrane | 211 | 636 |
| GrCBL9 | Gorai.008G255800 | 4.66 | 24.58 | −0.139 | Cell membrane | 213 | 642 |
| GrCBL10-1 | Gorai.010G101400 | 4.74 | 29.26 | −0.096 | Cell membrane | 252 | 759 |
| GrCBL10-2 | Gorai.009G045600 | 4.83 | 29.23 | −0.095 | Cell membrane | 253 | 762 |
| GhCBL1-1 | Gh_A11G0257 | 4.72 | 24.44 | −0.148 | Cell membrane | 213 | 642 |
| GhCBL1-2 | Gh_D11G0276 | 4.79 | 24.38 | −0.145 | Cell membrane | 213 | 642 |
| GhCBL1-3 | Gh_A03G0043 | 4.98 | 22.76 | −0.163 | Cell membrane | 199 | 600 |
| GhCBL1-4 | Gh_D09G1875 | 5.06 | 25.69 | −0.194 | Cell membrane | 224 | 675 |
| GhCBL1-5 | Gh_A09G1766 | 5.51 | 23.23 | −0.165 | Cell membrane | 200 | 603 |
| GhCBL3-1 | Gh_A01G0740 | 4.77 | 25.98 | −0.189 | Cell membrane | 226 | 681 |
| GhCBL3-2 | Gh_D01G0760 | 4.77 | 25.99 | −0.189 | Cell membrane | 226 | 681 |
| GhCBL3-3 | Gh_A13G1099 | 4.84 | 23.25 | −0.21 | Cell membrane | 202 | 609 |
| GhCBL3-4 | Gh_D13G1364 | 4.98 | 21.64 | −0.205 | Cell membrane | 189 | 570 |
| GhCBL3-5 | Gh_A04G0051 | 5.14 | 21.76 | −0.274 | Cell membrane | 189 | 570 |
| GhCBL3-6 | Gh_D05G3682 | 8.05 | 150.21 | −0.284 | Nucleus | 1,326 | 3,981 |
| GhCBL4-1 | Gh_A11G0126 | 4.77 | 23.01 | −0.059 | Cell membrane | 201 | 606 |
| GhCBL4-2 | Gh_D11G0140 | 4.82 | 24.97 | −0.185 | Cell membrane | 220 | 663 |
| GhCBL4-3 | Gh_A12G2144 | 4.97 | 24.5 | −0.175 | Cell membrane | 215 | 648 |

*(continued on next page)*

**Table 3** (*continued*)

| Gene name | Gene ID | pI | MW (kDa) | Hydrophilicity | Predicted subcellular localization | Amino acid residues | Coding sequence |
|---|---|---|---|---|---|---|---|
| GhCBL4-4 | Gh_A09G1696 | 5.27 | 28.4 | −0.184 | Cell membrane | 248 | 747 |
| GhCBL4-5 | Gh_D12G2320 | 5.64 | 25.06 | 0.023 | Cell membrane | 218 | 657 |
| GhCBL8 | Gh_D09G1801 | 4.85 | 24.74 | −0.177 | Cell membrane | 217 | 654 |
| GhCBL9 | Gh_D08G1764 | 4.74 | 23.8 | −0.032 | Cell membrane | 209 | 630 |
| GhCBL10-1 | Gh_A06G0800 | 5.18 | 33.56 | −0.143 | Cell membrane | 293 | 882 |
| GhCBL10-2 | Gh_D06G0922 | 4.95 | 30.41 | −0.159 | Cell membrane | 265 | 798 |
| GhCBL10-3 | Gh_A05G0335 | 5.16 | 30.48 | −0.114 | Cell membrane | 262 | 789 |
| GhCBL10-4 | Gh_D05G0440 | 5.01 | 30.25 | −0.08 | Cell membrane | 262 | 789 |

## RESULTS

### Genome-wide identification of the CBL family in two progenitor diploid and the tetraploid cotton species

The CBL genes in *Gossypium* were identified using the homologous alignment method. A total of 13, 13, and 22 CBL genes were respectively detected in A genome (*G. arboretum*), D genome (*G. raimondii*) and $A_tD_t$ genome (*G. hirsutum*) using 10 *Arabidopsis* CBL protein sequences as queries (Table 3). Further, the CBL candidate genes in *Gossypium* were confirmed by domain analysis programs of Pfam and SMART. The CBL family members were named according to their orthologous similarity to the 10 *Arabidopsis* CBL proteins (*Mohanta et al., 2015*). In general, the *CBLs* in *G. arboretum*, *G. raimondii* and *G. hirsutum* were named *GaCBLs, GrCBLs* and *GhCBLs*, respectively.

Most CBLs had very similar physical properties in the 3 *Gossypium* plants (Table 3). The open reading frame (ORF) lengths of the CBL genes ranged from 570 bp to 882 bp except that of *GhCBL3-6*, whose ORF length was 3981 bp. The GaCBL and GrCBL proteins contained 199-279 and 209-253 amino acids (AA), respectively, while GhCBLs were composed of 189-293 AA except GhCBL3-6, which consisted of 1326 AA. The molecular weights (MWs) of GaCBLs varied from 23.25 kDa (GaCBL10-1) to 32.43 kDa (GaCBL10-2), and of GrCBLs ranged from 23.25 kDa (GrCBL3-3) to 29.26 kDa (GrCBL10-1). The sizes of GhCBLs were 21.64 kDa (GhCBL3-4) to 33.56 kDa (GhCBL10-1) with an exception of GhCBL3-6 (150.21 kDa). The theoretical isoelectric point (pI) is small for overwhelming majority of the CBLs, ranging from 4.65 (GaCBL9) to 5.64 (GhCBL4-5). By contrast, pI of GhCBL3-6 was 8.05 (Table 3).

Putative subcellular localizations of the *Gossypium* CBL proteins were also analyzed. It was predicted that all of CBLs were located in cell membrane except that GhCBL3-6 was in the nucleus (Table 3). The quite different characteristics of GhCBL3-6 from other members suggest that GhCBL3-6 likely plays a special role in cotton.

### Distribution of the *Gossypium* CBL family members in the whole genome

Chromosomal distributions of the *CBL* genes were examined in *Gossypium*. In general, the *CBLs* were unevenly distributed among the *Gossypium* chromosomes. Thirteen *GaCBLs* were distributed on seven chromosomes. Among them, three *GaCBLs* were located on each

of Gachr07 and Gachr11 chromosomes. Two *GaCBLs* were situated in each of Gachr06 and Gachr13, and 1 *GaCBL* was on Gachr01, Gachr08 and Gachr09, respectively (Fig. 1). Thirteen *GrCBL* genes were identified on nine chromosomes. Each of the four chromosomes Grchr06, Grchr07, Grchr08 and Grchr09 owned 2 genes, and other chromosomes (Grchr02, Grchr03, Grchr04, Grchr10, Grchr13) individually contained one gene (Fig. 1). Likewise, 22 *GhCBL* family members were mapped onto 17 chromosomes. Each of the five chromosomes Ghchr09, Ghchr11, Ghchr19, Ghchr21 and Ghchr23 had two *CBL* members, and other chromosomes individually carried one *CBL* member (Fig. 1). We observed the phenomena of 2 CBL genes joining together in a chromosome. For instance, GaCBL4-2 and GaCBL9 were mapped within 16.0 Mb in Gachr06, and GrCBL4-3 and GrCBL9 were mapped within 53.8 Mb in Grchr08. These findings suggest that tandem duplication plays a role in generating these genes during evolution.

## Phylogenetic analysis and structural properties of *CBL* genes in *Gossypium*

To determine the sequence similarity relationship of the CBLs among *G. arboreum*, *G. raimondii*, and *G. hirsutum*, the phylogenetic tree for the 48 CBLs was constructed. The CBLs can be classified into four families (I to IV) (Fig. 2A). Family I consisted of 12 CBLs (three GaCBLs, three GrCBLs and six GhCBLs). The members in family II were 8 CBLs (two GaCBLs, two GrCBLs and four GhCBLs). Family III contained 14 CBLs (four GaCBLs, four GrCBLs and six GhCBLs). Family IV had 14 CBLs (four GaCBLs, four GrCBLs and six GhCBLs) (Fig. 2A).

The structure of a protein is closely related to its functions in cells. We therefore identified the intron-exon structures of the *CBL* genes in *Gossypium* by mapping the cDNA sequences onto their genomic sequences. Most of *GaCBLs* and *GrCBLs* owned eight exons except that *GaCBL3-2*, *GrCBL10-1*, *GrCBL10-2* had nine and *GaCBL9*, *GrCBL1-2* had seven. The majority of *GhCBLs* carried 7–11 exons, but *GhCBL4-4* had three exons and *GhCBL3-6* had 22 exons (Fig. 2A).

The putative domains in the *Gossypium* CBL proteins were also investigated. EF-hand motifs, which bind to $Ca^{2+}$ ions to transfer calcium signals, were observed in all CBL members. Each CBL proteins had three EF-hand motifs except for GaCBL9, which contained two such motifs (Fig. 2A). Furthermore, a conserved myristoylation motif (MGCXXS/T) was detected in the N-terminal regions of 11 CBL proteins. These proteins included four GaCBLs, two GrCBLs and five GhCBLs (Fig. 2B, 2C). A conserved palmitoylation site with N-terminal Cys residue at third, fourth, fifth or sixth position in amino acid sequence also existed in many cotton CBL members. The two sites are important in the attachment of a protein to membrane (*Mohanta et al., 2015*).

## Synteny analysis of CBL genes in *Gossypium*

To investigate the genetic origins and evolution of the CBLs in *Gossypium*, the homologous gene pairs among the *CBLs* from *G. arboretum*, *G. raimondii* and *G. hirsutum* were monitored, and the collinear analysis was carried out. The results revealed that 10 homologous gene pairs existed between *G. arboreum* and *G. hirsutum*, and 11 homologous

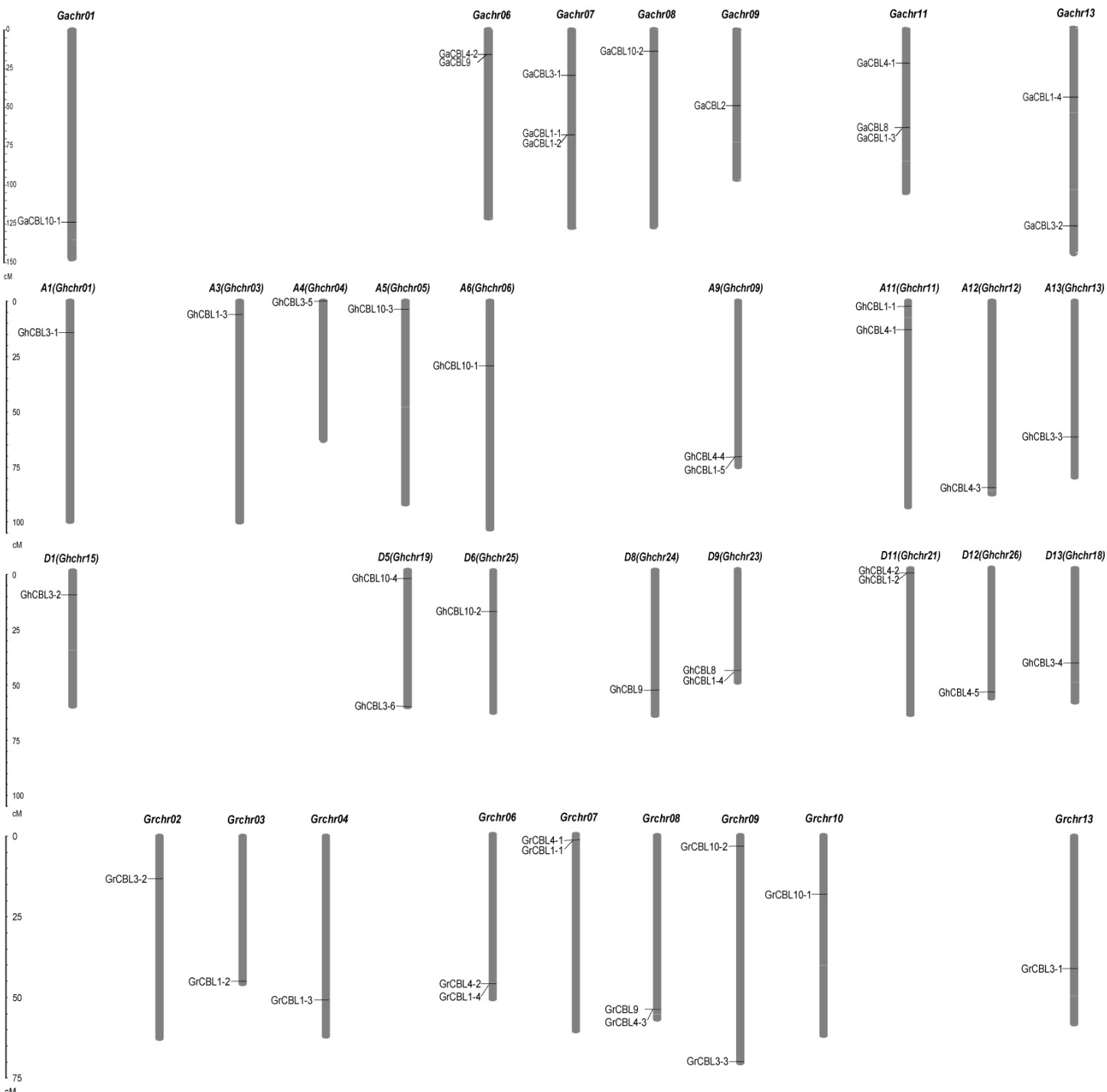

**Figure 1** **Distributions of the *CBL* family genes on chromosomes in *Gossypium*.** The *GaCBLs*, *GrCBLs* and *GhCBLs* are from *G. arboreum*, *G. rai-mondii* and *G. hirsutum*, respectively.

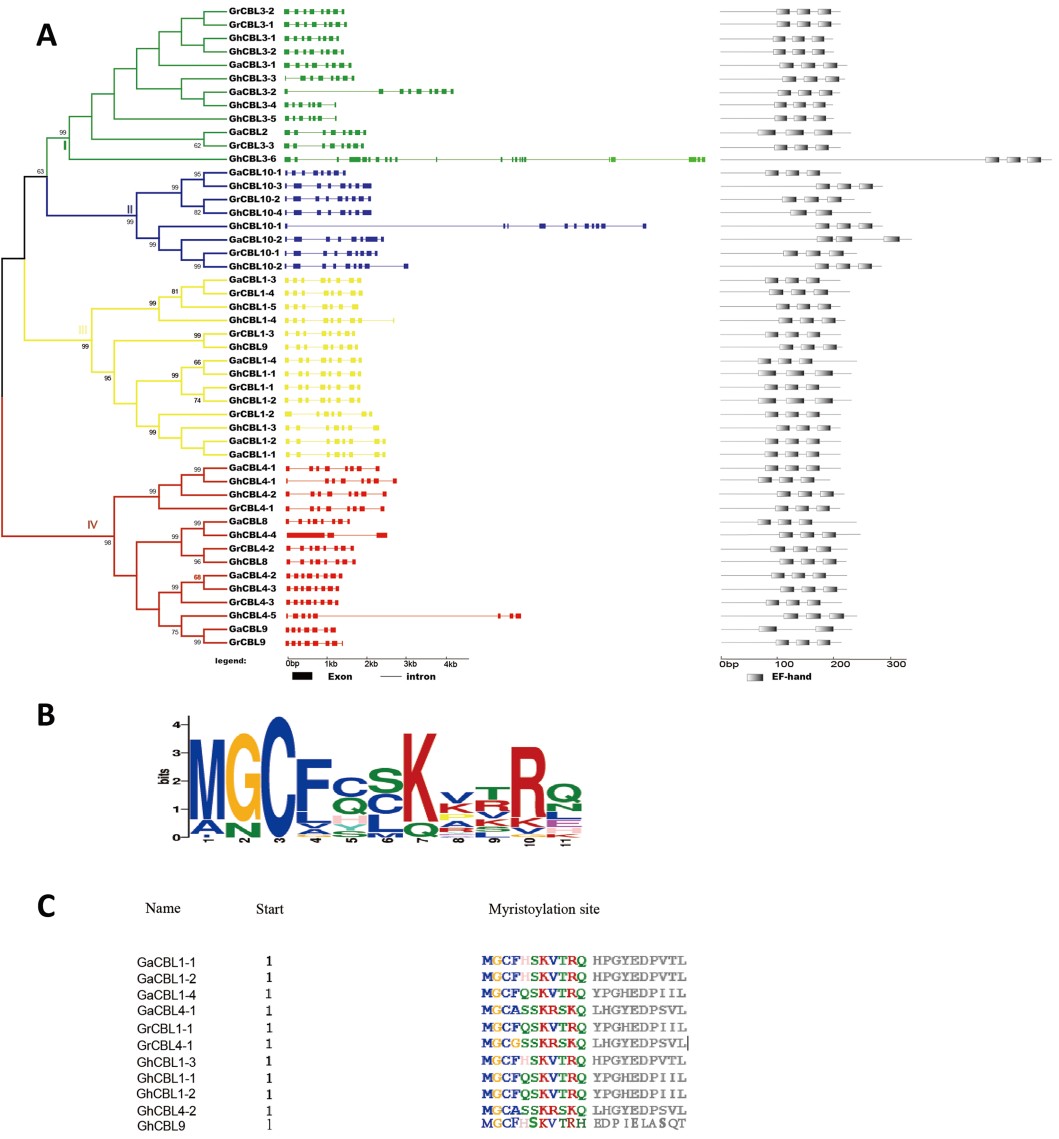

**Figure 2 Analysis of phylogenetic relationship, gene architecture and conserved domains of CBLs in Gossypium.** (A) The phylogenetic tree, exon-intron architecture and EF-hand domains of CBLs in *G. arboreum*, *G. raimondii* and *G. hirsutum*. The four major subfamilies are numbered I–IV. The color boxes indicate exons, and the color lines indicate introns; (B) The logo of the myristoylation motif. The capital letters stand for the amino acids, the higher the letter, the higher the conservation; (C) Multiple sequences containing the myristoylation motif in *Gossypium* CBLs.

gene pairs were found between *G. raimondii* and *G. Hirsutum* (Fig. 3A). Using the same method, seven homologous gene pairs were observed between *G. arboreum* and *G. raimondii*. They were distributed on five chromosomes in *G. arboreum* and five chromosomes in *G. raimondii*, respectively (Fig. 3B). Moreover, 212 homologous gene pairs (both based on orthology and paralogy) were found among the CBLs from the three *Gossypium* species (Table S1). These results imply that many cotton CBL genes may have evolved through segmental duplication.

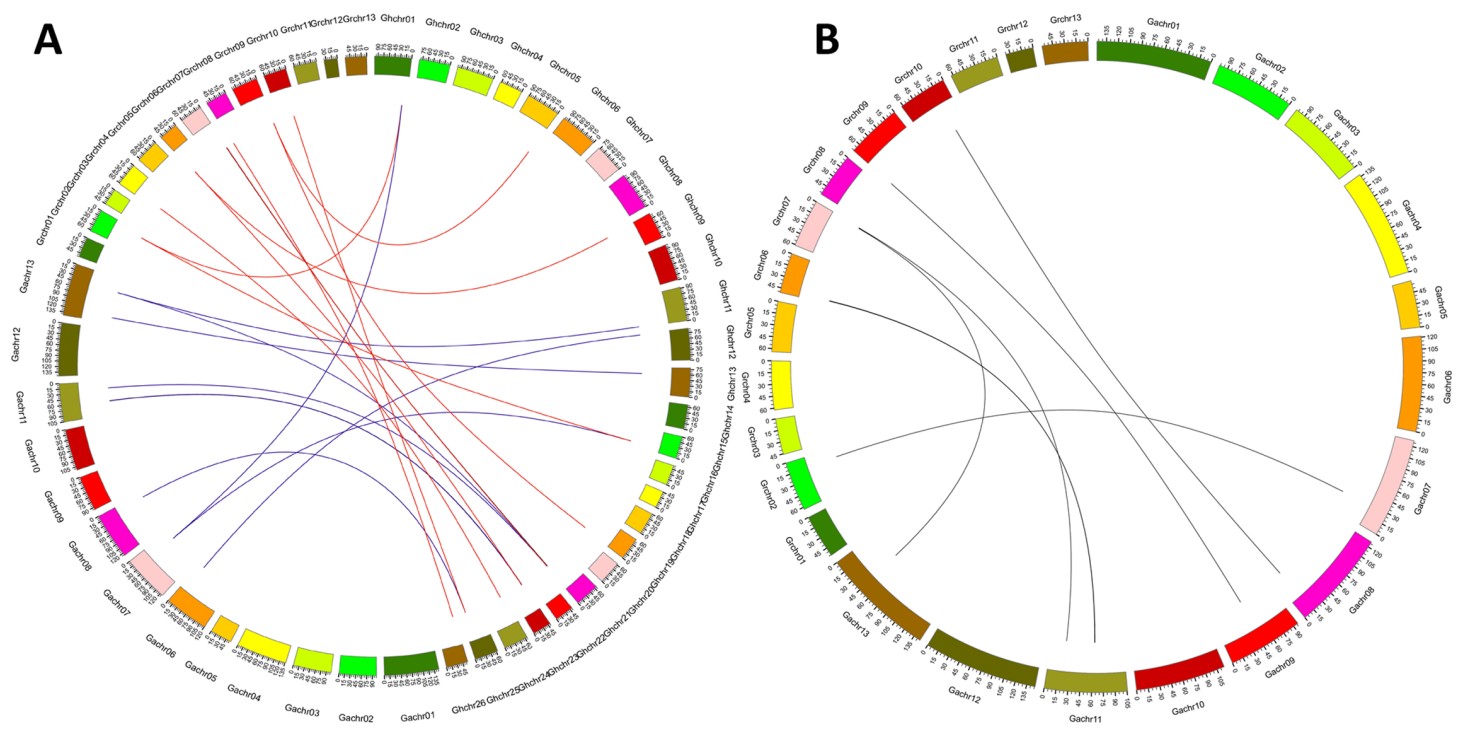

**Figure 3** **Genome-wide synteny analysis of *Gossypium* CBL genes.** (A) Synteny analysis between *G. hirsutum* and two diploid species *G. arboreum* and *G. raimondii*. Blue lines link gene pairs between *G. arboreum* and *G. hirsutum*, and red lines link gene pairs between *G. raimondii* and *G. hirsutum*; (B) Synteny analysis between *G. arboreum* and *G. raimondi*.

## Analysis of Ka/Ks values of the CBLs

To better understand the divergence of the *Gossypium CBL* genes after polyploidization, the value Ka and Ks and their ratio (Ka/Ks) were evaluated for the homologous gene pairs among *G. arboreum*, *G. raimondii* and *G. hirsutum* (Fig. 4, Table S2). The results showed that the Ka/Ks values among most of the homologous genes were less than 1, indicating they evolved under the purifying selection effect. Only GhCBL10-2/GrCBL10-1 has a Ka/Ks ratio more than 1, hinting that the gene pair may have been generated via the directional selection.

## Phylogenetic relationship of CBLs in *Gossypium* and other plant species

To gain insight into the evolutionary relationships among GaCBLs, GrCBLs, GhCBLs and CBLs of other plant species, we constructed a phylogenetic tree. Full-length amino acid sequences of 126 predicted CBL proteins were obtained from *G. arboretum*, *G. raimondii*, *G. hirsutum*, *A. thaliana*, *C. papaya*, *G. max*, *V. vinifera*, *T. cacao*, *P. trichocarpa*, *R. communis* and *O. sativa*. Phylogenetic trees were generated using the neighbor-joining method and MEGA 5.0 software. The CBLs family was divided into thirteen subfamilies according to the topology of the phylogenetic tree (Fig. 5). As expected, the three *Gossypium* CBLs commonly clustered closely in a subfamily. Most of them belonged to subfamily two, eight and thirteen. We found that the CBL members from different dicotyledon species

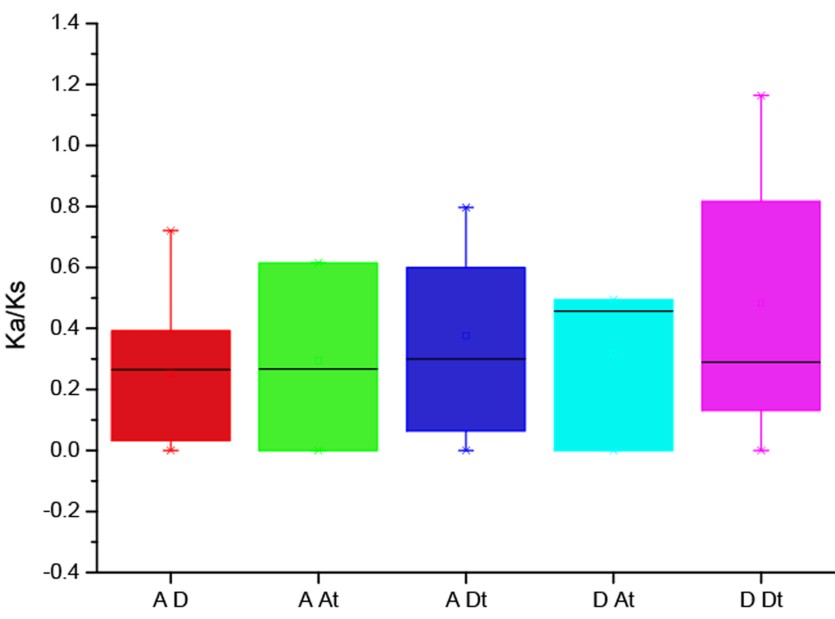

**Figure 4** The Ka/Ks values of the CBL homologous genes between the A genome, D genome and subgenomes of *G. hirsutum* ($A_t D_t$).

and rice always clustered in a subfamily, suggesting that the CBLs shared an ancestral sequence before the divergence of eudicots and monocots or convergent evolution events for these CBLs might have occurred in adaptations to drastic changes in the environment. Moreover, the CBLs from *Gossypium* plants often clustered together with those from *T. cacao* (Fig. 5). These results are expected because both *Gossypium* and *T. cacao* are in the *Malvaceae* family.

## Annotation analysis of GhCBLs

Putative functions of GhCBLs were analyzed using KOG (EuKaryotic orthologous groups (KOG) database (ftp://ftp.ncbi.nih.gov/pub/COG/KOG). Only the information on GhCBL3-6 was obtained. It was predicted that GhCBL3-6 played roles in modulation of RNA processing and modification, signal transduction, and coenzyme transport and metabolism. Gene ontology (GO) database for the 22 GhCBLs was also assessed. The result showed that these GhCBL members were capable of binding calcium ion, like those of other plant species. These analyses indicate that GhCBLs and other CBLs are of great importance in $Ca^{2+}$ signal transduction in plants.

## Expression analysis of *GhCBL* genes in tissues

The expression patterns of all the 22 *GhCBL* genes in tissues were monitored by qRT-PCR. We found that most genes were highly expressed in flowers except that *GhCBL4-3*, *GhCBL4-4*, and *GhCBL8* were dominantly expressed in roots and *GhCBL3-6* strongly expressed in leaves. Moreover, the transcripts of *GhCBL1-1*, *GhCBL1-4*, *GhCBL1-5*, *GhCBL3-4*, *GhCBL3-5*, *GhCBL3-6* and *GhCBL9* were relatively abundant in fiber, and those of *GhCBL4-3* were also numerous in flowers (Fig. 6). These results suggest that *GhCBL4-3*, *GhCBL4-4* and *GhCBL8* may mainly function in roots, *GhCBL3-6* mainly

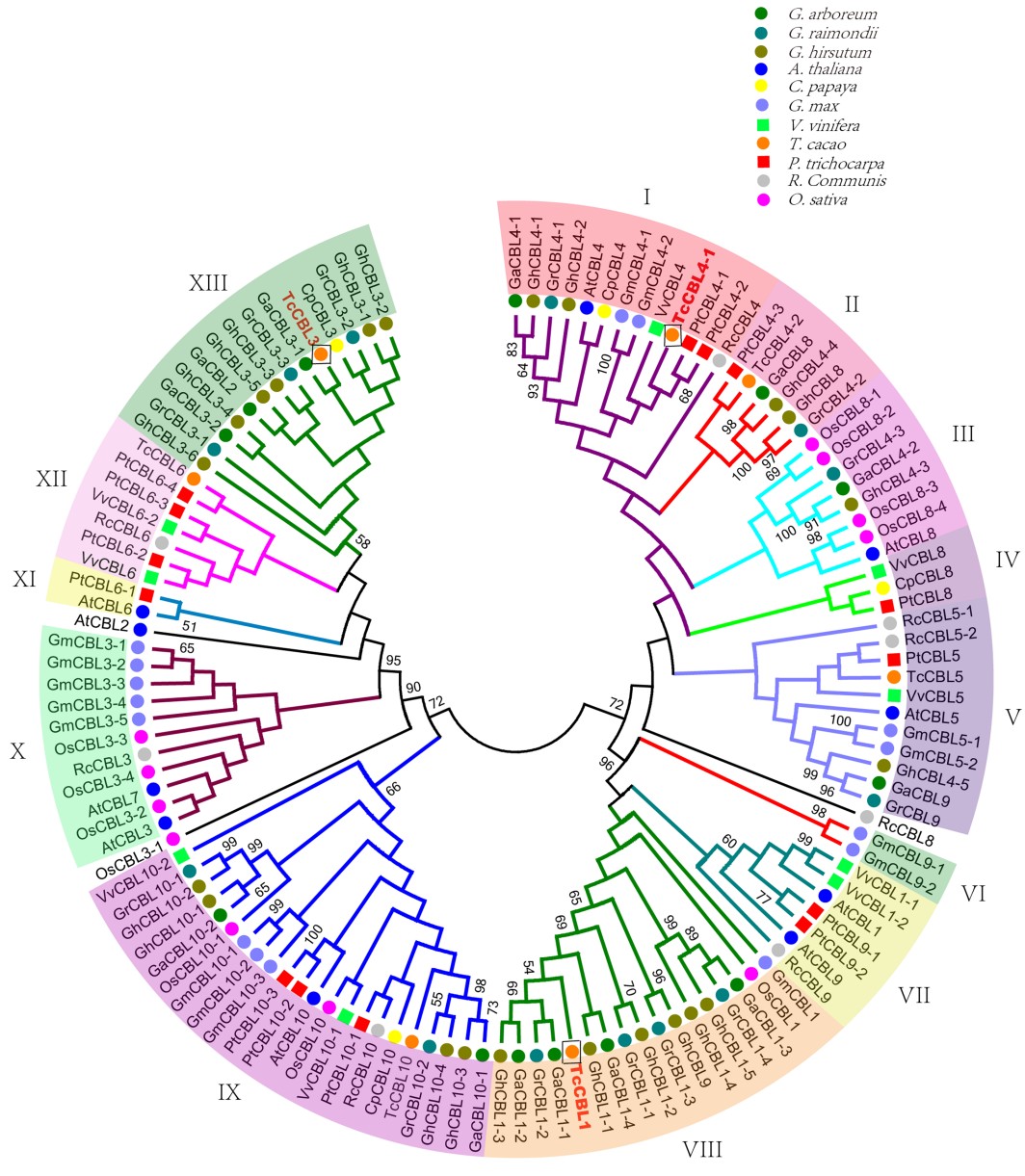

**Figure 5** **Phylogenetic tree of CBLs in *Gossypium* and other plant species.** The plants in the square frame indicate that the CBL genes outside of *Gossypium* have the closest evolutionary relationship with *Gossypium* CBLs.

functions in leaves and other genes may chiefly act in flowers. *GhCBL1-1*, *GhCBL1-4*, *GhCBL1-5*, *GhCBL3-4*, *GhCBL3-5*, *GhCBL3-6* and *GhCBL9* also probably play a part in fiber development in cotton.

## Expression patterns of *GhCBLs* in responding to potassium deficiency

CBLs have been addressed to play key roles in response to $K^+$ deprivation in *Arabidopsis* and rice (*Li et al., 2014a*; *Mao et al., 2016*). Accordingly, we measured the expression patterns of

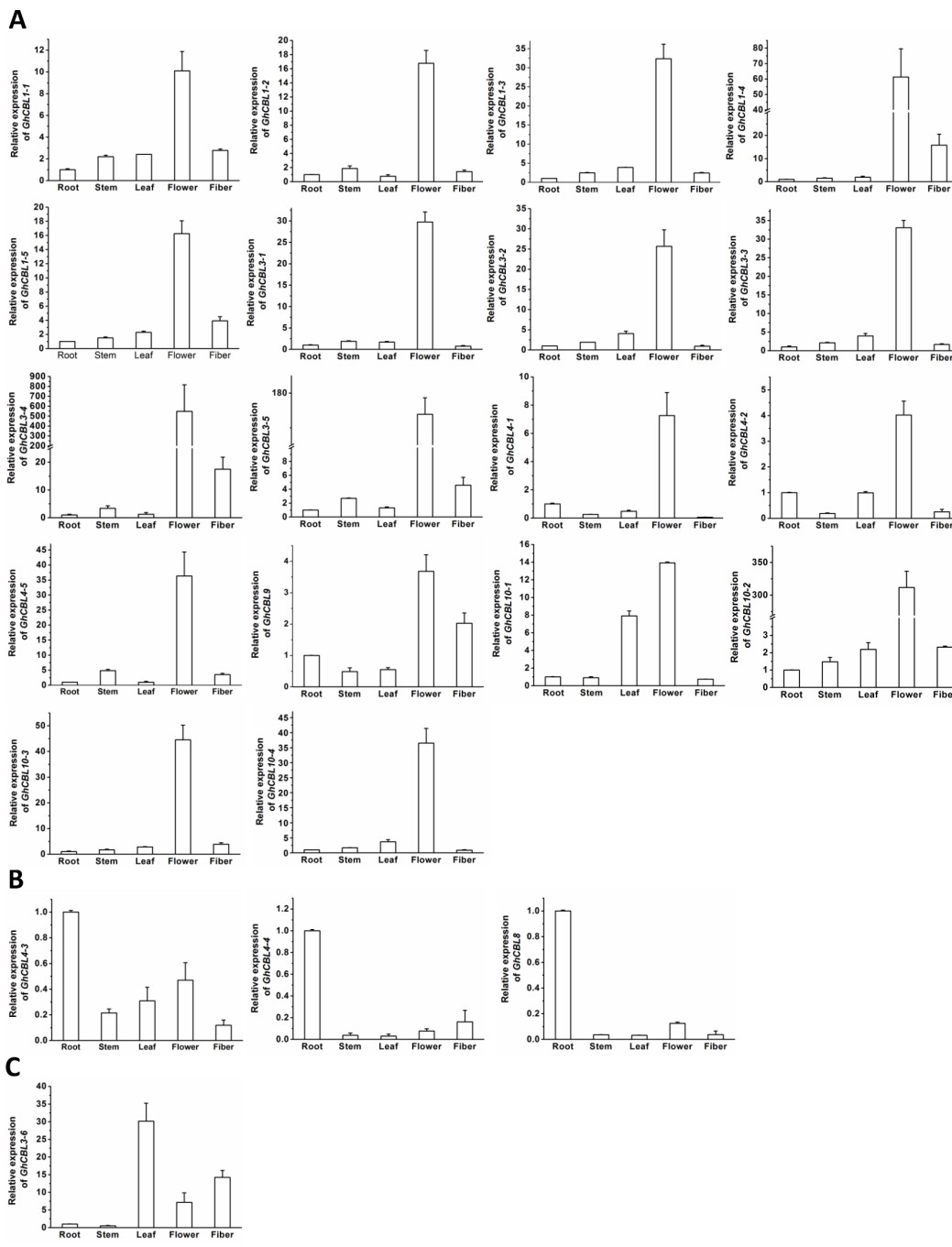

**Figure 6** **Expression of 22 *GhCBL* genes in tissues of cotton.** The genes preferentially expressed in flowers (A), roots (B) and leaves (C) are shown. The relative expression of genes was calculated from 3 independent replicates. The expression value of the gene in roots was set as 1. The vertical bars represent the standard error.

the 22 *GhCBL* genes in response to potassium deficiency. As a whole, potassium deficiency moderately altered the expression levels of *GhCBL* genes (Fig. 7). Under potassium deficiency, the transcripts of many genes were reduced at 6 h, but increased at 2 d and/or 5 d. These gene included *GhCBL3-1*, *GhCBL3-2*, *GhCBL3-3*, *GhCBL3-4*, *GhCBL4-4*, and *GhCBL10-3*. The expression levels of *GhCBL3-5*, *GhCBL3-6*, *GhCBL4-3*, *GhCBL4-5*, *GhCBL8* and *GhCBL9* were decreased while those of other genes were unchanged after shortage of potassium (Fig. 7). The effects of $K^+$ resupply on the abundances of *GhCBL* transcripts were also investigated. Compared with 5 d of low-$K^+$ treatments, 3 h of $K^+$ refeeding clearly resulted in decreases in the expression of many genes such as *GhCBL1-3*, *GhCBL1-5*, *GhCBL3-2*, *GhCBL3-3*, *GhCBL3-4*, *GhCBL10-1* and *GhCBL10-3*. However, $K^+$ resupply increased the expression of *GhCBL4-1*. The transcriptional levels of other genes did not significantly alter upon $K^+$ resupply (Fig. 7). These results suggest that a number of GhCBLs may play roles in response to potassium starvation in cotton.

### Several GhCBLs can interact with GhCIPK23 *in vitro*

To examine whether GhCBLs interact with GhCIPK23, yeast two-hybrid experiments were performed and total of 12 GhCBLs were measured. Among them, GhCBL1-2, GhCBL1-3, GhCBL4-4, GhCBL8, GhCBL9 and GhCBL10-3 were observed to interact with GhCIPK23. Furthermore, GhCBL1-2 and GhCBL9, the respective homologues of *Arabidopsis* CBL1 and CBL9, displayed more strong interactive signals with GhCIPK23 in yeast, suggesting that GhCBL1-2 and GhCBL9 may directly regulate GhCIPK23 in cotton.

## DISCUSSION

In the present study, we identified 13, 13 and 22 *CBL* genes in *G. arboreum*, *G. raimondii* and *G. hirsutum* genomes, respectively (Table 3). Among the 22 *GhCBL* genes, 11 and 11 were assigned to the $A_t$ and $D_t$ subgenome, respectively. They were similar to the number of *CBL* s found in *G. arboreum* and *G. raimondii*, respectively. We detected that eight *GaCBLs* and nine *GrCBLs* were homologous genes of *GhCBLs*. However, homologues of five GaCBLs and four GrCBLs were not discovered in the genome of *G. hirsutum*. These findings indicate that the eight *GaCBLs* and nine *GrCBLs* have been maintained in *G. hirsutum* after polyploidization event, while the five *GaCBLs* and four *GrCBLs* diverged from their orthologs in *G. hirsutum* during evolution. Moreover, we observed five *GhCBLs* (*GhCBL1-3*, *GhCBL3-5*, *GhCBL4-1*, *GhCBL4-4*, *GhCBL10-1*) in $A_t$ subgenome and two *GhCBLs* (*GhCBL3-4*, *GhCBL3-6*) in $D_t$ had no homologues in A genome of *G. arboreum* and D genome of *G. raimondii*, respectively. It is conceivable because selection pressures in diploids per loci are different than in the allotetraploid. Relaxed selection allows for development of novel and new functional alleles, but may also accumulate non functional, both at a higher rate possible that within the diploids. *G. arboreum* originates in the Africa/Arabia while *G. raimondii* and *G. hirsutum* originate in the Americas (*Wendel, Brubaker & Seelanan, 2010*). They are distributed in quite different places during evolution. Moreover, *G. arboreum* and *G. hirsutum* are two domasticated species (*Wendel, Brubaker & Seelanan, 2010*). Hence, geographic separation of the three species, and human selection may be essential for the diversity of the CBLs in *Gossypium*.

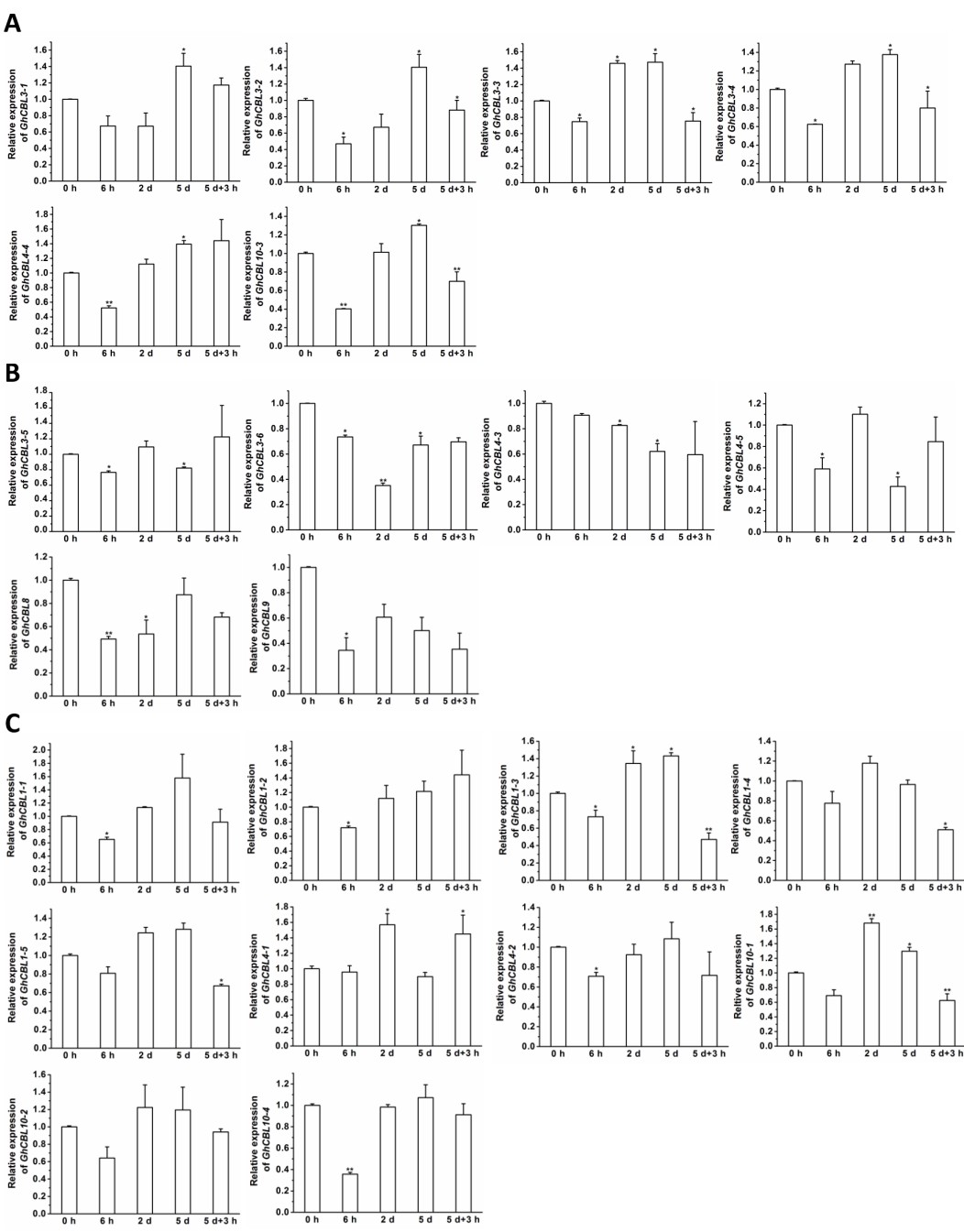

**Figure 7** **Expression of 22 *GhCBL* genes under potassium deprivation.** The relative expression of *GhCBLs* was examined under potassium deficiency or resupply for indicated period of time. Under K deficiency, the expression levels of the genes were decreased at 6 h but increased at 2 d and/or 5 d (A), were reduced (B), and were not altered (C) significantly. The expression value of the gene at 0 h was set as 1. The vertical bars mean the standard error. Statistical analyses were conducted by student's *t* test to assess the differences between the samples at 0 h and those at 6 h, 2 d, or 5 d as well as between the samples at 5 d and those upon resupplying potassium for 3 h (5 d + 3 h). The single and double asterisks means that the differences are significant ($P \leq 0.05$) and extremely significant ($P \leq 0.01$), respectively.

The physical properties of most GaCBLs and GrCBLs were similar to those of GhCBLs (Table 3), suggesting that the functions of the CBLs from the three cotton species remain highly conserved during evolution. The majority of *Gossypium* CBLs was predicted to localize in the membrane, just like many CBLs in *Arabidopsis* and rice. In *Arabidopsis*, CBL1 and CBL9 were described to localize in the PM. CBL2, CBL3 and CBL6 localize in tonoplast whereas CBL10 is in both PM and tonoplast (*Mao et al., 2016*). Rice CBL1 is also present in PM. The localizations of the CBLs should be consistent with their primary roles of sensing and transferring $Ca^{2+}$ signals in *Gossypium*. However, GhCBL3-6 was predicted to be nuclear. Its roles are unknown at present. Experimental characterization of GhCBL3-6 might shed light on some novel functions of it. GhCBL3-6 also gives obvious proof of the evolutionary advantage of being tetraploid. It may be a product of significant human intervention because nothing like it was seen in either diploid.

Analysis of gene distributions on chromosomes showed that most homologues of *GaCBLs* and *GrCBLs* in *G. hirsutum* were present in their corresponding $A_t$ and $D_t$ homologous chromosomes, respectively. These findings indicate that *GhCBLs* originate from DNA polyploidization. However, some *GhCBLs* homologues of *GaCBLs* and *GrCBLs* did not appear on their corresponding $A_t$ or $D_t$ chromosomes, suggesting that complex exchange events of chromosome segments occurred in *G. hirsutum* during evolution. Additionally, separated (e.g., *GaCBL4-1* and *GaCBL4-2*; *GrCBL1-1* and *GrCBL1-2*) and jointed (*GaCBL4-2* and *GaCBL9*) distributions of the *Gossypium* CBL homologous genes in chromosomes in combination with the colinearity results of these genes (Figs 1; 3) imply that both segmental duplication and tandem duplication are essential for the generation of cotton CBLs during genetic evolution. The number of introns in coding region of most CBL genes in *Gossypium* was six or seven, very similar to that in CBLs genes in *Arabidopsis*, rice, maize, wheat, canola and eggplant (*Kolukisaoglu et al., 2004*; *Zhang et al., 2014*; *Sun et al., 2015*; *Li et al., 2016a*; *Li et al., 2016b*; *Zhang et al., 2016*), reflecting the rather conserved structure of CBL genes in different species. Moreover, nearly all of the *Gossypium* CBLs shared three conserved EF hand domains with other higher plants (Fig. 2). In addition, many *CBLs* from *Gossypium* contained the myristoylated and palmitoylated sites, which may facilitate the targeting of CBL-CIPK complex to membrane. These features are also similar to those in *Arabidopsis*, rice and other plants (*Kolukisaoglu et al., 2004*; *Mohanta et al., 2015*). The conserved structure of these CBL family members in different plants might reflect a very similar mode of action and/or conserved interaction with their target protein CIPKs (*Mohanta et al., 2015*).

Measurement of the ratio of Ka to Ks indicated that majority of *Gossypium* CBL homologous genes have undergone purifying selection whereas *GhCBL10-2/GrCBL10-1* has experienced directional selection after polyploidization (Fig. 4). These results suggest that most *GhCBLs* have very high similarity in gene sequences and highly conserved functions to their orthologs from *G. arboretum* and *G. raimondii* during evolution. By contrast, a large divergence between *GhCBL10-2* and *GrCBL10-1G* has happened. *GhCBL10-2* may have evolved some novel functions through natural selection and human selection.

Phylogenetic analysis results revealed that the CBLs in *Gossypium* have closer relationship with those in cocoa than in other plants tested (Fig. 5). These findings strongly suggest that

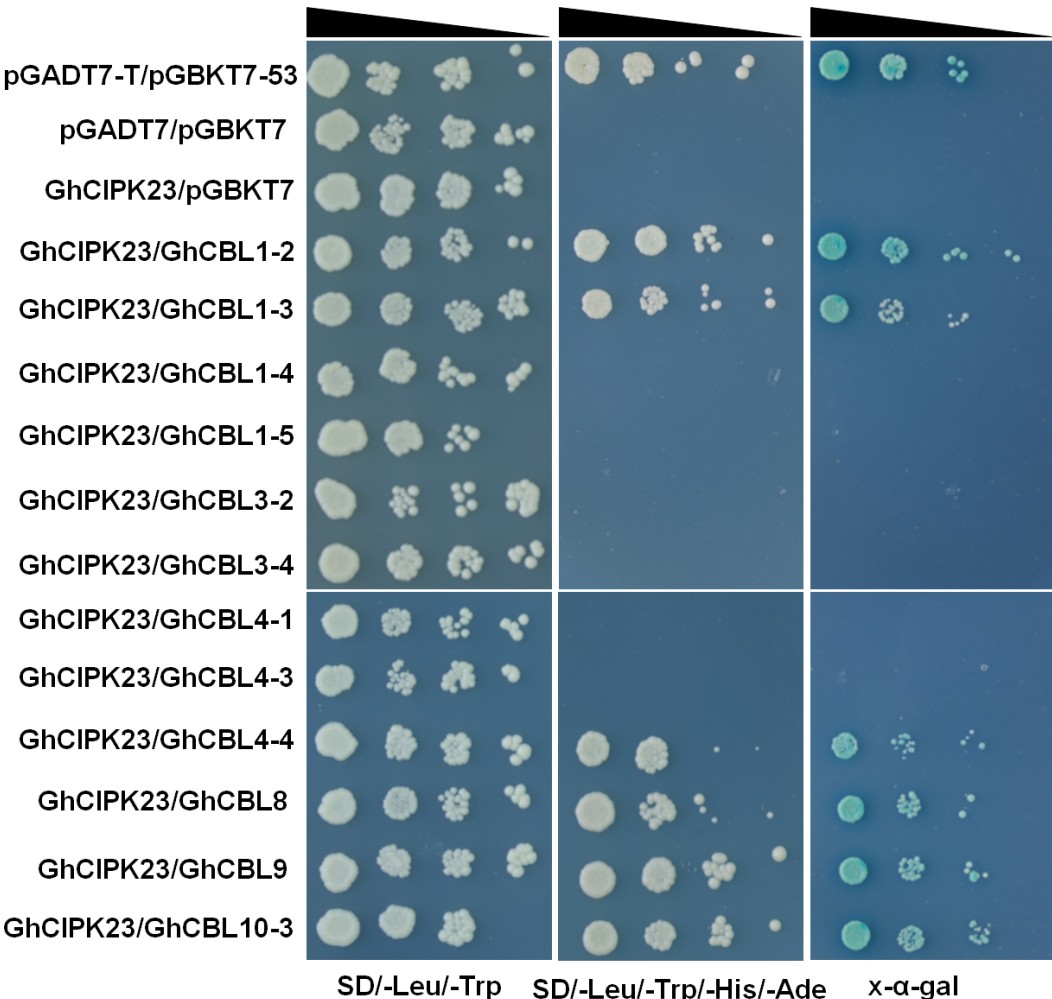

**Figure 8** **Yeast two-hybrid analysis of interactions between GhCBLs and GhCIPK23.** The yeast cells containing the indicated plasmids were grown on the non-selective SD/-Leu/-Trp solid medium and selective SD/-Leu/-Trp/-His/-Ade solid medium, followed by X- α-Gal staining. The reduced cell densities in the dilution series are shown by narrowing triangles when proceeding from left to right. The first row represents a positive control, the 2th and 3th rows represent two negative controls.

the cotton species may have a more recent common ancestor with cacao relative to other plant species, in line with the results of other gene families in *Gossypium* (*Li et al., 2014b*; *Li et al., 2016a*; *Li et al., 2016b*). It may justify using CBL as another evolutionary model in plants because it showed highest similarity with another taxon from the same family and may help to narrow down the most vital or evolutionarily conserved or ancient sequences in *Gossypium*.

Expression analysis results showed that almost all of the *GhCBL* genes were expressed in various tissues including the root, stem, leaf, flower and fiber. Of note, most genes were dominantly expressed in the flower and fiber (Fig. 6), hinting that these genes may play important roles in the reproductive development in cotton. *G. hirsutum* is a highly domesticated plant for its seed fiber, which is developed from the flower. Preferential

expression of many *GhCBLs* in flowers and fibers suggests that human selection markedly affects the genetic variation and expression profiles of *GhCBLs*. Besides, the expression levels of *GhCBL4-3 GhCBL4-4* and *GhCBL8* in roots were clearly higher than those of other genes. These data imply that the three genes may function in modulation of ion transport or acclimation to diverse abiotic stresses in roots. Their detailed actions and mechanisms will be examined in the future.

The expression of 22 *GhCBLs* in responding to potassium starvation was determined. The transcription of most genes was moderately promoted at 2 d and/or 5 d post low-potassium treatments (Fig. 7), indicating multiple GhCBL genes likely regulate cotton response to potassium deprivation. Strikingly, in *Arabidopsis*, the expression of *CBL1* and *CBL9* was reported to be stable, and the transcripts of *CBL10* in roots were moderately decreased under low-potassium conditions (Cheong et al., 2007; Ren et al., 2013). These results imply that constitutive expression of some *CBL* genes may be enough for transmitting $Ca^{2+}$ signals to downstream targets in response to potassium deficiency in plants. Thus, those *GhCBLs* that were not induced by low-potassium stress also likely play a part in adaptations to potassium deprivation in cotton. However, which sequences and how GhCBLs regulate potassium starved responses remains to be investigated in the future.

CIPK23 has been observed to function in diverse cellular processes in *Arabidopsis* (Mao et al., 2016). In this study, six out of 12 GhCBLs could interact with GhCIPK23 in yeast (Fig. 8), indicating that different GhCBL members may interact with and modulate GhCIPK23 in various growth and/or stress responses in cotton. The cotton homologues of *Arabidopsis* CBL1 and CBL9 suggest that GhCBL1 and GhCBL9 probably play similar roles to CBL1 and CBL9 in cotton.

### Funding

This work was supported by the Science and Technology Development Program of He'nan in China (No. 162102110005) and Foundation of He'nan Educational Committee of China (No. 15A210018, No. 17A180018 and No. 14B180029). The funders had no role in study design, data collection and analysis, decision to publish, or preparation of the manuscript.

### Grant Disclosures

The following grant information was disclosed by the authors:
Science and Technology Development Program of He'nan in China: 162102110005.
Foundation of He'nan Educational Committee of China: 15A210018, 17A180018, 14B180029.

### Competing Interests

The authors declare there are no competing interests.

### Author Contributions

- Tingting Lu conceived and designed the experiments, analyzed the data, wrote the paper, prepared figures and/or tables, reviewed drafts of the paper.

- Gaofeng Zhang performed the experiments, wrote the paper, prepared figures and/or tables, reviewed drafts of the paper.
- Lirong Sun analyzed the data, contributed reagents/materials/analysis tools, technical support.
- Ji Wang performed the experiments.
- Fushun Hao conceived and designed the experiments, analyzed the data, wrote the paper, reviewed drafts of the paper.

## Data Availability

The raw data has been uploaded as a Supplemental File.

## Supplemental Information

Supplemental information for this article can be found online at http://dx.doi.org/10.7717/peerj.3653#supplemental-information.

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
