# Peer review of "Genome-wide identification of CBL family and expression analysis of CBLs in response to potassium deficiency in cotton"

_PeerJ, doi:10.7717/peerj.3653_

## Round 0.1 · original submission · Major Revisions

In addition to addressing the reviewer comments, please do not use the terms "up-regulated" and "down-regulated" when what is meant is higher or lower expression levels.

·

Basic reporting

Distinguish the use of cotton (normally reserved for G. hirsutum) as a general term for all three species in your study. This is confusing to the readers that normally associate cotton with just the taxon G. hirsutum (and G. barbadense, and even diploids such as G. arboreum and G. herbaceum).
Some of the software listed has no source and/or reference. Please insert them (e.g. lines 113-144). Give the reader some indication of how you went from protein sequences of Arabidopsis CBLs to the genome (nucleotide sequences) of the three Gossypium species. There is no indication of how you identified the CBL domains, just that you used Pfam and HMMER software and again no reference is listed. I understand there is a lot of experimental work here but details are still needed for the reader or at least a reference of the procedures to another article. Otherwise split up some of the work into separate articles. Lines 145-146 mention harvesting fibers from ovules but this is a delicate procedure but no details on age of ovules, how the fibers were removed (I assume in a sterile laminar flow hood) and further processed. Reverse transcriptase reactions are just a few lines in your materials and methods. If the manufacturer provided instructions then give the product number and page numbers of the instructions that came with the product.
Give more details on the Yeast Protocols Handbook. Is this a product from Clontech (catalog number) and what chapter/page numbers have the methods? What is the SD media? Did you buy this premade or make it yourself and what amounts of each amino acid was added?
I insist figure 1 be rearranged with G. arboreum on top (as is), then first row of G. hirsutum (1-13), then the rest of G. hirsutum (14-26), then G. raimondii on the bottom, because I thought 1-13 of G. hirsutum was orthologous to G. arboreum, and 14-26 of G. hirstum was orthologous to G. raimondii? Envisioning this setup enables me to see some parallels of interesting discussion such as GrCBL3-1 is present on GRchr13 and should be collinear with Ghchr26 and there I see GhCBL4-5. You should emphasize that the geographic separation of these three species, chromosome location and even impact of human selection are additional evolutionary forces, not just sequence changes. Show the logic of your research, not just the data. This is my logic: if G. raimondii and G. arboreum both had the same CBL sequence or just one gene then G. hirsutum would just start with two CBL. Then all G. raimondii CBLs would cluster together and all of G. arboreum would cluster together. But the figure show that they do not, there are some interspersed. Therefore each species already contained more than one CBL prior to formulating G. hirsutum. Different CBLs in G. raimondii and G. arboreum reflect their independent evolutionary history because they are on different continents. If any are similar it is either convergent evolution or some hint of an ancestral sequence. Then G. hirsutum is much more complicated but we can hope to resolve some of it with other approaches like sub genome chromosome locations. The exceptions to finding similarity of A, D, to AtDt are the different evolution events of G. hirsutum offered by their polyploid genome and/or human selection. How do we hope to ever try to resolve the convergent evolution vs ancestral sequences still remaining? Maybe we look at function/structure which you did.
Figure 2b is confusing to me. It looks more like artwork than a figure. Put in an explanation for it that most readers can understand.
Figure 5 is ‘cool’ and I can navigate around it and clearly see clustering by species and family. Emphasize that T. cacao is in the same family Malvaceae, as cotton.

Experimental design

This is my first exposure to the journal so I will guess that the research is appropriate. It has a wide range of investigative tools to appeal to a wide audience.
I am satisfied with the multiple techniques used to answer multiple questions.
List more detail in materials and methods or cite the references or exact sources of your protocols.
This is the primary weakness of the article, lack of detail and/or references on the protocols. We have a wide scientific audience and a relatively new field of CBLs so the details are very important. You should want to give yourselves credit for the large amount of work in this study as it appeared to have resulted in many of your expected findings and validated your approaches.

Validity of the findings

It follows the literature well. It strengthens the body of work with BLAST, Arabidopsis, genome sequences, many tools online to analyze genetic/protein sequences. Even membrane functions were involved so it ties well to the wide body of knowledge in plant biology.
It should stress the importance that so many other layers of expression (protein, cellular location, expression up or down, etc.) were involved to study this trait that is heavily validates assigning function when only having genome sequence.
I would have gone much further with speculation, but that is just my opinion. Some indication that the nuclear CBL in G. hirsutum, their other CBLs sequences and their activity in fiber expression all suggest to me that they could be products of human selection and would be fair speculation. Future research to resolve them in the other tetraploid species would be very exciting and help to resolve this.

Additional comments

Get some references from Jonathan Wendel so we are all following the expert on Gossypium evolution. Your research will work very well with his findings but there is not one listing of his work.
I made corrections using the pdf by highlighting sections and in the 'note' typing in the correct text.
Comments are also placed in the pdf with the 'note'.

Reviewer 2 ·

Basic reporting

The authors have prefaced three experiments with five figures of cursory analysis of information that is already available elsewhere.
The title succinctly reveals this mash-up: "Genome-wide identification of CBL family AND expression
analysis of CBLs in response to potassium deficiency in cotton."

Figure 1 shows the locations of CBL genes in the draft reference genomes from Gossypium (cotton) species, which is available in the annotation files that accompany the reference genomes.

Figure 2 involved a small amount of analysis, perhaps a day, and can be directly reproduced only from files already released with the reference genomes. The significance of conserved exon structures is not discussed. The acquisition of new introns during speciation is not discussed.

Figure 3 essentially provides another visualization of the information in Figure 1.

Figure 4 can also be re-derived from genome sequences without much effort. The significance of directional selection of GhCBL10 is not explored.

Figure 5 re-presents the phylogeny of CBL genes from Fig 2, this time including orthologous sequences from eight non-cotton plant species.

Figure 6 does represent novel, biological work. Although much more exhaustively covered by RNAseq at CCnet (eg http://structuralbiology.cau.edu.cn/gossypium/gene_detail_AD.php?gene=Gh_D06G0922 ) these RTqPCR experiments are an independent observation of the expression of CBL genes in cotton, and should be published.

Figure 7 is the novel experiment that I suspect was the impetus for this project and manuscript. Since the level of induction or repression of CBL transcripts is less than two-fold during the time course of potassium deprivation observed, I further understand why this paper is padded with the descriptive structural genomics preface. However, as I understand the mission of PeerJ, I do consider these data publishable. It is worth knowing that CBL genes only have a limited response in cotton roots under the described conditions. The assertion that "many genes were down-regulated at 6h, but up-regulated a2d and/or 5 day," seems a stretch to me, but should be made more clear in the presentation of Fig 7. Include asterisks and p-values to show that bars are significantly different. Group the genes according to similar expression profiles.

Figure 8 is also novel work that should be published. Although the interaction data presented from these yeast two-hybrid experiments confirm, in cotton, the relationships of orthologs already established in Arabidopsis, the cotton community does benefit from these data. What about GhCBL10 ?

Discussion of "lost or regenerated" CBL genes following polyploidization and "newly evolved genes" are contradictory. If a CBL gene exists only in the allotetraploid G. hisutum, but neither G. arboreum or G. raimondii, it may have been lost from both extant diploids, rather than newly evolved. There are more careful ways to make these arguments.

The concluding statement is confusing: "Thus, those GhCBLs that were NOT upregulated by low-potassium stress likely participate in modulation of potassium absorption and/or transport." I would rather expect that potassium stress would affect the transcription of genes involved in potassium absorption. Please clarify and expand your reasoning.

Typo throughout: capitalization of genus eg “G. Hirsutum”

Experimental design

Figures 5-7 contain novel experiments.
Please see my comments above.

Validity of the findings

Figures 5-7 contain novel experiments and are presently clearly, although p-values and asterisks are needed for Figs 5 and 6.

---

## Round 0.2 · accepted · Accept

Thank you for addressing the reviewer comments.